# A Review of Converter Circuits for Ambient Micro Energy Harvesting

**DOI:** 10.3390/mi13122222

**Published:** 2022-12-14

**Authors:** Qian Lian, Peiqing Han, Niansong Mei

**Affiliations:** 1Shanghai Advanced Research Institute, Chinese Academy of Sciences, Shanghai 201210, China; 2University of Chinese Academy of Sciences, Beijing 100049, China; 33PEAK INCORPORATED, Shanghai 201203, China

**Keywords:** energy harvesting, charge pump, rectifier, maximum power point tracking

## Abstract

The Internet of Things (IoT) has a great number of sensor nodes distributed in different environments, and the traditional approach uses batteries to power these nodes: however, the resultant huge cost of battery replacement means that the battery-powered approach is not the optimal solution. Micro energy harvesting offers the possibility of self-powered sensor nodes. This paper provides an overview of energy harvesting technology, and describes the methods for extracting energy from various sources, including photovoltaic, thermoelectric, piezoelectric, and RF; in addition, the characteristics of the four types of energy sources and the applicable circuit structures are summarized. This paper gives the pros and cons of the circuits, and future directions. The design challenges are the efficiency and size of the circuit. MPPT, as an important method of improving the system efficiency, is also highlighted and compared.

## 1. Introduction

The demand for monitoring and management of smart homes, wearable devices, biomedical and industrial systems, and infrastructure has enabled the development of the Internet of Things (IoT) in recent years [1,2,3]. The IoT requires billions of wireless sensor network (WSN) nodes, which conventionally rely on battery power: however, the resultant problem is that the battery capacity is insufficient to last the entire sensor life cycle, and the labor and resources required to replace the widely distributed batteries require huge maintenance expense. In addition, the large size of the batteries, and environmental pollution, mean that this method of powering the IoT nodes is not the optimal solution. Consequently, technologies that harvest the ambient energy around sensors are gaining attention as power solutions for low-power IoT applications, with a view to replacing batteries for powering nodes.

The environmental energies that have been applied include solar, thermal, vibrational, and radio frequency (RF) energy, and they are obtained through photovoltaic cells (PVCs), thermoelectric generators (TEGs), piezoelectric harvesters (PEHs), and wireless energy harvesters (WEHs), respectively. The energy density of environmental energy sources, due to their fluctuating nature, is usually in the range of μW-mW/mm^2^, with the possibility of going down to the pW order of magnitude under extreme conditions: for example, in an indoor environment, PV cells can harvest energy in the range of 10–100 μW/mm^2^, while in an outdoor environment this value increases to 10–100 mW/mm^2^. For piezoelectric and thermal energy, the environment in which the PEH and the TEG are located will also have an impact on the energy density harvested: if they are based on human harvesting, the resonant frequency of the PEH is lower, and the range of energy that can be harvested by the TEG is reduced, with the result that the energy density of both is in the μW order; however, if the PEH and the TEG are in an industrial environment—where the former has a higher resonant frequency, and the latter can harvest an increased temperature difference—then the energy density can reach the order of mW. Furthermore, RF energy is too weak, and it is difficult to harvest energy up to the mW level, but the advantage is that the energy is uninterrupted. Table 1 classifies the four types of energy harvesters according to their characteristics:

The biggest problem with energy in the natural environment is its instability and weak nature. How to efficiently harvest environmental micro energy in a small area is the biggest challenge faced by the micro energy harvester system: for example, some energy sources have very low input voltage (<0.2 V), and conventional voltage converter systems suffer poor conversion efficiency or are even no longer valid; therefore, it is necessary to explore more effective boost circuits. This paper summarizes the respective characteristics and applicable topologies of the four energies, including the advantages and disadvantages of various boost converter structures and application scenarios—with special attention given to the applicability of different methods of maximum power extraction—and it also gives suggested future development directions. The rest of this article is organized as follows: Section 2 describes the circuit structures of the PV energy harvesting, and compares the pros and cons of the MPPT methods used in the micro energy harvesting scenario; Section 3 presents the differences between thermal energy harvesting and photovoltaic energy harvesting; in Section 4, the circuits applied to piezoelectric energy harvesting are introduced; Section 5 summarizes the structure of RF energy harvesting; in Section 6, the structures for simultaneous harvesting of multiple energies are described; Section 7 explains the directions and suggestions for future development; finally, Section 8 concludes the paper.

## 2. Photovoltaic Energy Harvesting

The benefit of solar energy is high density and ubiquity: the shortcoming is its nature, in that it is heavily influenced by the weather, and it is not readily available. Solar energy is generated via PVCs, to produce direct current, and a DC–DC converter transforms the unstable low voltage to a stable high voltage. Switching circuits, based on inductors or capacitors, can be used to implement DC–DC converters, with boost structures often used for inductor-based converters, and charge pump structures for capacitor-based converters. The capacitor-based converter is the main structure applied in solar energy collection systems, due to its ease of integration, low EMI, and flexible boost function. The output voltage of the converter can be regulated by a low dropout regulator. The unavailability of sunlight in dark environments makes the energy storage unit an integral part of the system, in order to maintain smooth operation. Assuming that the performance of the photovoltaic modules is in ideal conditions—such as adequate illumination and unshaded—then the following two factors will improve the efficiency of a solar energy harvesting system: (1) improving the efficiency of the DC–DC converter; (2) reducing the energy losses between the PV cells and the DC–DC converter, which can be achieved by the maximum power point tracking (MPPT) technique. However, when the power output is different between the PV cells, the generated power will be significantly restricted: this condition is known as “unbalanced generation”. The imbalance may come from various sources, such as shadows from obstacles, moving clouds, and dust covering. Xiao et al. reported that this mismatch can have a disproportionate impact on system performance, because the module cell with the lowest output limits the current path through the other elements in the string, which can produce significant power losses. Consequently, for multi-source solar energy harvesting systems it is necessary to research anti-shading function, to cope with the losses due to the power imbalance between cells, arising from shading by trees or objects, that tends to occur in the application scenarios [4]. In high-power solar systems, this mismatch is usually solved with a flyback topology, whereas in microscale energy harvesting systems, this approach is too large and lossy.

To solve the problem of unbalanced solar power generation in micro energy harvesting systems, Jeong et al. have proposed an efficiency-aware collaborative multi-charging system that can improve the total conversion efficiency by adaptively distributing the input power among different cells, according to the power level, when the PV cells are partially shaded [5]. Kermadi et al. have proposed an MPPT technique whereby the MPP tracking is guaranteed under complex partial shading conditions [6].

For semiconductors, the types of loss include conduction and switching. Switching losses are positively correlated with frequency; however, the efficiency of MPPT decreases when the frequency is low; therefore, there is a frequency at which a balance between reducing switching losses and increasing the MPPT efficiency can be achieved. At the same time, the conduction losses are positively correlated with the forward conduction resistance. The three losses contribute differently to the total losses at different power levels: hence, the frequency can be adjusted appropriately at various system powers to achieve the lowest overall losses [7].

### 2.1. Structure of the Inductor-Based Energy Converter

Liu et al. have applied a dual-path six-switch (2P6S) converter with boost structure, as shown in Figure 1a. The 2P6S converter includes two paths: the direct path and the indirect path; the charge reaches the load from the PV cell through the direct path; the indirect path is from the PV cell through the battery to the load [8]. Wang et al. have proposed a dual-path three-switch (2P3S) converter based on the 2P6S structure, as shown in Figure 1b. The 2P3S converter reduces the number of charge-sharing power switches, but increases the direct path efficiency, while decreasing the indirect path power conversion efficiency (PCE); it also has the limitation that *V_P_* and *V_B_* must be greater than *V_L_* [9]. To reduce the efficiency of the indirect path, Huang et al. have proposed a single-path three-switch (1P3S) converter, as shown in Figure 1c [10]. The disadvantages of the 1P3S converter are the lack of direct paths, and the constraint that *V_P_* and *V_L_* must be larger than *V_B_*. The three configurations are suitable for different situations: 2P6S is the universal solution for most applications; 2P3S is the solution for a WSN with frequent operation; and 1P3S can be used in a WSN with a low duty cycle and standby power.

### 2.2. Charge Pump Topologies

Inductor-based converters are usually suitable for applications larger than 100 mW. The disadvantage of inductor-based technology is that it typically requires an off-chip inductor in the mH level, occupying a space of about 100 mm^3^, whereas a surface mount ceramic capacitor can take up a size of fewer than 0.5 mm^3^. In addition, capacitor-based structures have the benefit of being chosen for low-power-IoT and small-area applications, because they are easy to integrate. Given the cost efficiency of using capacitors, capacitor-based technologies have been developing in recent years.

The charge pump (CP) is one of the inductor-less DC–DC converters. This capacitor-based boost circuit was first proposed by Dickson in 1976, and is widely used in integrated circuits [11,12]. The chain of the Dickson CP is shown in Figure 2: each cell includes a charge transfer switch (CTS) and a pump capacitor, where the CTS acts as a switch, and consists of a diode or transistor [13].

The main factor that limits the efficiency of the Dickson CP structure and the output voltage is the threshold voltage (*V_TH_*), which can be seen in Equation (1):(1)$$\begin{array}{c} V_{OUT}=\left(V_{IN}-V_{TH}\right)+N\left(\frac{V_{CK}}{1+\alpha_{T}}-V_{TH}\right)-N\frac{I_{OUT}}{f_{CK}\left(1+\alpha_{T}\right)C^{\prime }} \end{array}$$
where $V_{CK}$ is the voltage of the clock pulse, $\alpha_{T}$ is the ratio of the CTS parasitic capacitance to the pump capacitance, $I_{OUT}$ is the current at the load, and $f_{CK}$ is the clock frequency. To improve the CP’s conversion efficiency, novel techniques are proposed.

### 2.3. Efficiency-Improving Topologies

#### 2.3.1. Gate Biasing and Body Biasing Techniques

Controlled-switch technology using a CP was applied to memory for the first time. A simplified structure of the gate biasing technique is shown in Figure 3. In [14,15] it is used to generate positive and negative voltages efficiently: the difference is that NMOS is employed for positive generation, and PMOS is applied for negative generation. This scheme is characterized by the need to use a four-phase non-overlapping clock and an auxiliary boost capacitor. In this process, the gate voltage is constant, and is not affected by other voltage fluctuations, and *M_i_* can conduct, as long as the gate-to-source voltage is higher than the *V_TH_*.

As the MOS switch in a bootstrap CP can replace the MOS diode in a Dickson CP, the ON resistance can be significantly reduced; however, it is difficult for the bootstrap CP to continue to achieve high conversion efficiency when the input voltage *V_IN_* is below 100 mV. Fuketa et al. have improved the bootstrap CP, and have proposed a gate-boosted CP, where changing the signal amplitude of *φ_1B_* and *φ_2B_* from *V_IN_* to the output voltage *V_M_* can further reduce the ON resistance, and enhance performance [16].

The *V_TH_* is affected by the difference between the substrate and the source voltage; therefore, the bias of the wells of the MOSFET can be employed to reduce the *V_TH_* so that the effective input voltage of the CP is further reduced. The body biasing solution was first proposed by Sawada et al. in [17], as shown in Figure 4, where the source and the substrate of the MOSFET are shorted together. This connection allows the MOS to have the same threshold, and decreases the reverse losses.

#### 2.3.2. Cross-Coupled Charge Pump

The cross-coupled CP, also called the latched CP, evolved from a two-stage CP, shown in Figure 5a, and is usually used as a voltage doubler or shifter: it was first proposed by Nakagome et al. as a word line driver with feedback, and so the structure is also known as Nakagome’s cell [18]. Figure 5b shows the coupling of Nakagome’s cell with a double series MOSFET, which was first published by Gariboldi and Pulvirenti as a monolithic quad line driver, a structure that constitutes a unit of a cross-coupled CP [19].

Cross-coupled CPs, as shown in Figure 5c, are similar in structure to dual-branch circuits; therefore, the pumping capacitance of a single-stage cross-coupled CP is half that of a traditional Dickson CP, so the corresponding MOSFET size is smaller. The cross-coupled structure reduces circuit ripple, and increases charge transfer efficiency. In addition, gate biasing and body biasing techniques can be superimposed on the cross-coupled CP structure, to further improve the PCE. The MOSFET is in a linear condition in the operating state, which turns off when in the inverted state.

Based on the above, many literatures have proposed modification of the CP structure. In [20], the charge current of a three-stage cross-coupled CP without and with a phase-shifted clock was derived, and the experimental results showed a threefold reduction in the charge current ripple, and achieved a PCE of 69%. Tsuji et al. proposed a low-leakage current driver matched with a three-stage CP to achieve high output voltage, wide load current, and 70.3% peak efficiency [21]. Peng et al. utilized both gate biasing and body biasing techniques to operate the CP circuit in the subthreshold region, with operating voltages as low as 320 mV in a 0.18 μm standard process [22]. In addition, forward body biasing (FBB) was first proposed by Chen et al. in [23], and this technique, when applied to a three-stage CP circuit, can achieve an output current improvement rate of 150% at an input voltage of 0.18 V. The dynamic body biasing (DBB) technique employed in [24] enhances the current transfer level, by preventing the reverse current of the cross-coupled NMOS, resulting in a 240% increase in the maximum output current, compared to using only conventional FBB.

#### 2.3.3. Clock-Boosted Charge Pump

An additional improvement to the Dickson CP is the clock boost method. A quantitative analysis of the dynamic performance of the CP is presented in [25], which shows that decreasing the number of stages facilitates a reduction in rise time: however, this mechanism is at the expense of reducing the output voltage amplitude. The solution is to increase the clock voltage, which can reach twice the amplitude of the supply voltage by cascading Nakagome’s cells, as in Figure 6b. A CP circuit with a clock booster (CKB) can reduce the rise time while maintaining the same area, or cut the area of the silicon used while keeping the rise time constant, and the improvement rate can reach 60%; however, the limited drive capability of the CKB potentially weakens the speed benefits of the reduced number of stages.

Yi et al. proposed a differential bootstrapped ring-VCO (BTRO). The BTRO can harvest energy at low input voltages, to achieve a 1:10 boost ratio [26]. A BTRO composed of three inverter delay cells can generate a six-phase clock signal with swing boost, reducing the number of driven CP stages to only three, effectively increasing the PCE while significantly reducing the dependence of the clock frequency on the load capability. Each inverter delay unit that makes up the BTRO consists of two voltage triplers with two pairs of differential outputs to drive the delay unit and the CP element, respectively.

### 2.4. MPPT Techniques

The I/V characteristics of PVCs are nonlinear and have only one MPP, a value that varies with light level and temperature. To enable the circuit to continuously capture the MPP, and thus extract the maximum power, the MPPT technique was developed. The PV MPPT techniques are summarized in [27], including the hill climbing algorithm, the perturbation and observation (P&O) algorithm, conductance increment, fractional open-circuit voltage (FOCV), fractional short-circuit current (FSCC), and so on, to a total of 19 MPPT methods.

As the implementation of MPPT in micro energy harvesting systems needs to satisfy the characteristics of low cost, small size, and high efficiency, Han analyzed and compared the best fixed voltage (TBFV), FOCV, FSCC, load current maximization (LCM), and P&O methods that are applicable to micro systems. The comparative results show that the FOCV method is simple and inexpensive in circuitry, consumes less power, and can meet sufficient accuracy requirements, making it the most commonly used MPPT method in micro energy harvesting systems [28]. In addition, the P&O and hill climbing methods, as well as the artificial neural network (ANN) method, have been used extensively in solar energy harvesting.

#### 2.4.1. Principle of the FOCV Method

The $V_{MPP}$ of PV arrays are linearly related to $V_{OC}$ at different irradiance and temperature conditions, which is the principle of the FOCV method:(2)$$\begin{array}{c} V_{MPP}\approx \alpha V_{OC} \end{array}$$
where α is a constant, determined by the characteristics of the PV array, and takes a value between 0.71 and 0.78. The empirical value of α for a particular PV array needs to be determined in advance under different conditions, when being applied. The $V_{OC}$ is extracted by briefly disconnecting the power converter, and then passing it through the operator to reach the MPP: however, power losses are incurred in this process, and additional circuitry and memory devices—usually large capacitors—are required to maintain power to the load during the disconnection circuit. Various approaches have been proposed, to solve these problems. To reduce the losses in the disconnected circuit, Kobayashi et al. propose that the voltage generated by the PN junction diode can reach 75% of the $V_{OC}$, which can avoid arithmetic, and directly approximate the $V_{MPP}$ [29]. A delay block is added in [30], to reduce the sampling time, which can decrease the losses, and a flash ADC is used in [31], to quantize the $V_{OC}$ for the purpose of cancelling large capacitors.

Although Equation (2) is an asymptotic formula, it is adequate for approximation in cases where low precision is required: it does not demand complex controls and operations, and can make a good compromise between cost and performance.

#### 2.4.2. The Hill Climbing and P&O Algorithms

The hill climbing and P&O algorithms are two methods of performing MPPT optimization to produce the same effect. The hill climbing method provides perturbation, by changing the duty cycle of the power converter, while the P&O algorithm offers perturbation by altering the operating voltage of the PV array; both algorithms yield the same result when the converter and the PV array are connected [27]. With the gradual increase in the application of capacitive DC–DC converters for energy harvesting, research on the hill climbing and P&O methods is also increasing. Unlike FOCV, the hill climbing and P&O methods are applied without disconnecting the circuit. The on-chip implementation of the MPPT control system with the P&O method retains high performance characteristics with low design complexity [32].

To match the impedance of the boost converter with the PV cell, the switching frequency, switching width, conversion ratio, and capacitance value of the converter can be adjusted separately or simultaneously: this is called the hill climbing method. When only one variable is changed, it is called one-dimension (1-D) MPPT, and is used in [33,34]; two-dimension (2-D) MPPT [33] changes two variables; three-dimension (3-D) MPPT is used in [35] to track the MPP of the PV cell by adjusting three variables. The advantage of the hill climbing method is high accuracy, and the disadvantages are a complex control circuit, high power consumption, and slow convergence; another problem is that the maximum power extraction cannot be effectively performed when the environment is changing rapidly and the PV module is shaded, as with the P&O method.

The power consumption of the FOCV, P&O, and hill climbing methods is presented in [36]: the data show that these three MPPT methods consume energy in the order of µW.

#### 2.4.3. Other Promising MPPT Control Methods

The negative feedback control (NFC)-based low-overhead adaptive MPPT (AMPPT) method was first proposed by Lu et al. in [37], with the benefit that it has a simple control circuit and low power consumption compared to the hill climbing method: however, the disadvantage is that it requires the design of a dedicated voltage-controlled oscillator (VCO), so that it is not suitable for non-custom cases. In order to improve the adaptability of the circuit system to PV cells, and to make the system more suitable for industrialization, Wang et al. have proposed an improved AMPPT method aided by a neural network (NN) model [38]. Unlike the electrical equivalent model traditionally used to simulate PV cells, the neural network-assisted adaptive MPPT (NN-A-AMPPT) uses data to train the NN model to simulate PV cells, and proposes a reconfigurable VCO based on BJT.

The application of artificial intelligence (AI) algorithms, such as NN and optimization, as well as hybrid algorithms for solar energy harvesting, is described and compared in [39], which includes ANN and particle swarm optimization (PSO), for accurately tracking MPPs in any atmospheric conditions. Yap et al. evaluated the pros and cons, unresolved issues, and technical implementations of AI-based MPPT techniques [40]. For low-power systems, AI algorithms require large amounts of data, with slow convergence and high power consumption. Three NN algorithms corresponding to solar energy harvesting were analyzed and compared in [41]. Ahmed et al. compared the effectiveness of four traditional MPPT algorithms and ANN algorithms, and the results showed that the artificial intelligence network had the best MPP tracking [42]. Tabrizi et al. have proposed a digital MPPT algorithm that can achieve maximum power transfer by varying the number of CP stages under different load conditions when using a CP as a DC converter, and can avoid disconnecting the circuit from the source, suitable for ultra-low power consumption [43].

## 3. Thermal Energy Harvesting

Similar to solar energy, thermal energy is also in the form of DC, but the energy harvesting based on thermal energy is more sensitive to losses. In addition, of the four energy sources mentioned in this paper, the driving ability of the circuit based on thermal energy harvesting is the strongest. Unlike PV cells, the simple series configuration of thermoelectric modules can extract most of the available power, even in the presence of rather large mismatches among the modules. In addition, the MPPT method applied to solar energy is suited to thermal energy. FOCV is the most commonly used MPPT method. The results of four MPPT algorithms—P&O, Incremental Conductance (INC), Open Circuit Voltage (OCV), and Short Circuit Current (SCC)—applied to the same thermoelectric energy harvesting system were analyzed in [44], and show that the OCV algorithm has the fastest charging speed; however, unlike solar energy harvesting, α is taken as 0.5 in the FOCV based on thermal energy harvesting.

Thermal energy is characterized by low harvesting voltage: the output voltage of TEGs that harvest thermal energy is only in the range of 40 to 100 mV/K [45], so a boost converter with a low starting voltage is needed, and because of the low starting voltage, converters with self-starting capabilities are desired.

A fully built-in low-voltage start-up boost converter was proposed in [46], which can achieve a minimum start-up voltage of 82 mV. The input voltage range is 60–460 mV, and the output voltage range is 1–1.8 V. It has the static power of microwatt level, and a peak efficiency of 78.55%, by using a modified Schmitt trigger low-leakage logic gate. Radin et al. co-designed the oscillator and CP to provide self-starting and efficient conversion at ultra-low input voltages, achieving a minimum start-up voltage of 11 mV, and an operating voltage of 7.3 mV, with a peak efficiency of 85% at 140 mV input voltage, enabling the autonomous and efficient operation of the on-body device provided by a TEG at a temperature in the order of 1 °C gradient [47].

As thermal energy is more sensitive to losses, zero-current switching (ZCS) technology is commonly used in converters, to further reduce system energy losses and improve efficiency [7,30,47]. ZCS is a soft-switching technology used to ensure that the current waveforms do not overlap during switch-on and switch-off, thereby greatly reducing switching losses, as opposed to hard-switching technology, which has higher switching losses [48].

## 4. Piezoelectric Energy Harvesting

A complete PEH system can be divided into a vibration source and an interface circuit, in which the vibration source consists of a current source, *I_p_*, an intrinsic capacitance, *C_p_*_,_ and a resistance, *R*, in parallel. *R* can be neglected sometimes, as the value of *R* is much higher than *C_p_* at the resonant frequency. Ambient vibration sources typically have frequencies below 500 Hz: as a result, an interface circuit is needed to convert AC power to DC power, as required by the WSN. PEH interfaces can commonly be divided into three categories: (1) energy storage device-free; (2) inductor-based; (3) capacitor-based.

### 4.1. Energy Storage Device-Free Interface

An energy storage device-free interface comprises a rectifier with a simple structure that is easy to implement for full interface integration. The rectifier is dominated by a full-bridge rectifier (FBR), which sets a high *V_TH_* for the generated energy extracted by the circuit. Figure 7 shows the FBR circuit used for piezoelectric energy harvesting, and its corresponding waveform, respectively. The shaded part on the right shows the losses caused by the voltage flip, from which it can be seen that it significantly limits the power efficiency, especially at low excitation levels.

Ghovanloo et al. have proposed a fully integrated wideband high-current rectifier, replacing the two diodes in the FBR by two cross-coupled PMOS transistors. A comparison between the FBR and the PMOS rectifier with cross-coupling is shown in Figure 8 [49]. The gates of the cross-coupled PMOS transistors have a larger voltage swing drive than the diode-connected transistors, so a higher on/off current ratio can be achieved: however, the bottom two diodes are still implemented by the diode-connected transistors to block the reverse current, and the efficiency is not optimized [50].

To further reduce the reverse current, an active rectifier based on PMOS with cross-coupling is proposed, to improve the efficiency. The power losses due to the voltage drop of the forward voltage can be greatly reduced by replacing the diode with actively controlled transistors. Chang et al. have proposed a rectifier with active and non-overlapping control, as shown in Figure 9: the unbalanced comparator reduces the reverse current, and the non-overlapping control minimizes the additional losses caused by the transistors turning on and oscillating at the same time, and improves the power extraction efficiency and the voltage conversion ratio [51]. In this way, a PCE of 95% can be achieved at 20 kΩ and 200 kΩ load conditions.

### 4.2. Inductor-Based Interface

Inductor-based rectification techniques include synchronous electrical charge extraction (SECE) [52], energy investing (EI) [53], and synchronous switch harvesting on inductor (SSHI) [54]. In order to improve the power extraction efficiency and reduce the losses due to the FBR, a phase-controlled switch and an inductor can be added to the FBR to form an SSHI circuit, with the degree of performance improvement defined by the maximum output improvement rate (MOPIR):(3)$$\begin{array}{c} MOPIR=\frac{P_{rect,max}}{P_{FBR,max}} \end{array}$$
where $P_{rect,max}$ and $P_{FBR,max}$ are the maximum output powers of the corresponding interface circuits at the rectifier and FBR outputs, respectively.

Figure 10 shows the structure of the SSHI circuit and its waveform: the synchronous pulse signal $\phi_{SSHI}$ controls *L*, and thus the *RLC* oscillation loop, to flip the voltage. The value of $V_{PT}$ is $V_{S}+2V_{D}$ or—($V_{S}+2V_{D}$) before $I_{P}$ crosses the zero point, and the voltage flips to −$V_{F}$ or $V_{F}$ after the zero point. The absolute value of the resulting flip voltage, $V_{F}$, is always less than $V_{S}+2V_{D}$, due to the resistive damping. $V_{F}$ can be expressed as $V_{F}=\left(V_{S}+2V_{D}\right)\exp\left(-\pi /\sqrt{\left(4L/\left(R^{2}C_{P}\right)\right)-1}\right)$ [55]. After the voltage flip, the value of |*V_F_*| continues to rise to $V_{S}+2V_{D}$, during which the energy losses of the shaded part are generated. It can be seen that the losses generated by the SSHI circuit are greatly reduced, compared to the FBR. The efficiency of the SSHI circuit can be presented as:(4)$$\begin{array}{c} \eta_{SSHI}=\frac{V_{F}}{V_{S}+2V_{D}}=exp(-\frac{\pi }{\sqrt{\frac{4L}{R^{2}C_{P}}-1}}) \end{array}$$
where *C_P_* is the intrinsic capacitance of the piezoelectric transducer (PT), *L* is the inductor, and *R* is the total resistance in the *RLC* loop, including the DC resistance of the inductor, the on-resistance of the switch, and other parasitic components.

Improving the performance of the SSHI circuit can be achieved by increasing the value of *L* and decreasing the value of *C_P_* and *R*. It is worth noting that increasing *L* will result in a certain growth in *R*. The design of the SSHI in [54] can improve the energy extraction performance by an MOPIR of 5.8, with a DC output power of 40.6 µW and a power density of 8.12 mW/cm^3^.

SSHIs can be divided into series synchronous switch harvesting on inductors (S-SSHIs), and parallel synchronous switch harvesting on inductors (P-SSHIs). The SSHI synchronously flips the voltage on the PT, to minimize energy wastage due to internal capacitor charging and discharging, making it one of the most energy-efficient circuits, which ideally has no charge wastage; however, the efficiency of the SSHI does not reach 100%, due to the synchronous switching damping effect. Dell’Anna et al. have summarized the characteristics of other inductor-based rectifier schemes, such as SECE, Hybrid SSHI, and EI [56].

### 4.3. Capacitor-Based Interface

Capacitor-based circuits applied to piezoelectric energy harvesting include the synchronized switch harvesting on capacitor (SSHC) [57], the flipping-capacitor rectifier (FCR) [58], and the split-phase flipping-capacitor rectifier (SPFCR) [59], which allow on-chip integration by avoiding the use of off-chip inductors. 

Figure 11 shows the circuit diagram of the SSHC interface with one charge-swap capacitor and its corresponding waveform. A switched-capacitor (SC) interface circuit uses three pulsed signals ($\phi_{0}$, $\phi_{n}$, and $\phi_{P}$) to control five switches without overlapping. When the current crosses zero, and $V_{PT}$ needs to be flipped from the high voltage $V_{S}+2V_{D}$, then $\phi_{P}$, $\phi_{0}$, and $\phi_{n}$ conduct in turn, to bring $V_{PT}$ down to −1/3 ($V_{S}+2V_{D}$) and vice versa, with a voltage flipping efficiency of 1/3. The voltage flipping efficiency further increases when more SC modules are added. Du et al. has summarized the voltage flipping efficiency for an SSHC with 1–8 SCs, and the inductance required for an SSHI to achieve the same efficiency, respectively, which shows that an SSHC can significantly reduce the system size by using capacitors [55].

Capacitor rectification needs to achieve more flipping phases with less capacitors, to make the energy extraction efficiency as high as possible; however, too many flipping phases increase the control losses; therefore, the flipping phases need to be minimized without sacrificing the energy extraction efficiency. The SPFCR is proposed in [59], to achieve 21 flipping phases with four capacitors and MOPIR up to 9.3 times.

### 4.4. Maximum Power Extraction

Maximum power transfer occurs when the input impedance of the AC–DC converter is a complex conjugate of the PT source impedance; however, matching the capacitive component of the PT source with an inductor in the converter is impractical, as the inductance can be very large—in the tens to hundreds of the Henry range [60]. Li et al. proposed an up-conversion technique, to reduce the system area by impedance matching, after shifting the frequency of the energy generated by the PT source to a higher value [61]. On the other hand, because the source impedance varies with the excitation frequency, the load needs to operate at optimal conditions, in order to improve efficiency. This means that the DC–DC converter also needs to convert the rectified voltage according to the load requirements.

Ottman et al. used a DC–DC converter followed by the rectifier to regulate the load, but the circuit used two stages [62]. John Turner et al. used a bidirectional DC–DC converter with feedforward control to simulate a parallel RL load, but the circuit power dissipation was relatively large for small-scale systems [63]. Chew et al. implemented simultaneous MPPT and voltage regulation, using a single DC–DC converter [64]. The SSHI technique eliminates the capacitance term by voltage flipping, but does not reach the maximum power transfer. SSHIs and SSHCs can harvest energy efficiently from broadband excitation: however, the final power extracted is all load-dependent; hence, the variation of the load affects the efficiency.

A series of MPPT methods have been proposed. In [65,66] an FBR combined with the FOCV method, to extract the maximum power from the PT, was applied. To improve the rectifier efficiency, Kawai et al. designed a P-SSHI circuit with FOCV [67]. The FOCV method requires disconnecting the circuit to detect the open-circuit voltage: to reduce the power losses, Fang et al. proposed a method called fractional normal operating voltage (FNOV), based on S-SSHI, to avoid disconnecting the PEH from the load [68]. P-SSHI circuits combined with P&O MPPT techniques were proposed by Li et al. in [69], and a self-adjusting phase-shift-based SECE technique was proposed by Morel et al. in [70]. Wang et al. proposed an emerging MPPT technique based on envelope extraction without programmable control units and without disconnecting the circuit, with rectifiers using SSHI architecture [71]. An adjustable-delay SSHC method was proposed by Liao Wu et al. in [72], to regulate the power flowing into the load, by adjusting the pulse width of the switching pulse signal in the SSHC, avoiding the use of a second-stage circuit and inductors, and thus reducing the complexity of the circuit. In [59], a switched-capacitor DC–DC converter was composed of capacitors in the non-flipping state for load regulation, and a multi-voltage conversion ratio (MVCR) was obtained. An active rectifier circuit with two-dimensional P&O control was proposed in [73], which could track the maximum power point quickly, and be error-free.

## 5. RF Energy Harvesting

Compared to piezoelectric, RF energy has lower power density, yet the advantage is that it is not affected by energy irregularities, and has high reliability. The RF energy harvesting (RFEH) system can be classified into near-field RFEH and far-field RFEH: when the distance between transmitter and receiver is within the Fraunhofer distance, it is called near-field RFEH, and vice versa for far-field RFEH. Far-field RF is also known as ambient RF. The power density of far-field RF is lower than that of near-field RF.

Unlike near-field RF, far-field RFEH in the electric field (E) and magnetic field (H) has equal amplitude in the same point: hence, the received power of the antenna is quantifiable and predictable, and the received power can be calculated by
(5)$$\begin{array}{c} P_{RX}=\frac{P_{TX}G_{TX}G_{RX}\lambda^{2}}{{\left(4\pi R\right)}^{2}} \end{array}$$
where *P_TX_* is the transmitted power, *G_TX_* is the transmitting antenna gain, *G_RX_* is the receiving antenna gain, and *R* is the distance between the transmitting and receiving antenna. The architecture of the RFEH system is shown in Figure 12.

As the rectifier is a nonlinear circuit, input impedance will vary with the level of *V_in,rec_*. It is necessary to design the impedance matching network (IMN) to ensure matching between the output impedance of the antenna and the input impedance of the rectifier. In RFEH applications, the IMN has another role, of reducing the reflection from the load to the input; in addition, the IMN sometimes acts as a passive booster to improve the sensitivity of the circuit. The IMN consists of inductors and capacitors in general. Co-design, transformer matching, and reconfigurable matching can also be applied to IMN design [74].

The DC–DC converter can assist in boosting the voltage when the DC voltage output from the rectifier is not enough to power the load. The DC–DC converter also has the role of regulating the load for impedance matching, and is an optional module in RFEH systems. As this DC–DC converter with higher input voltage is not used as the main converter, and the technology is mature, it is not introduced in this paper.

### 5.1. RF–DC Rectifier

Converters that transform RF signals are called RF–DC, and are measured in terms of power metrics, including sensitivity and PCE. Factors that affect both measures are the input and output signal, the circuit topology, and the device parameters. RF–DC rectifiers are divided into three categories: (1) diode rectification; (2) Dickson method; (3) cross-coupled rectification [74].

Schottky diodes have a low turn-on voltage characteristic [75], and can replace diodes, to improve converter efficiency; however, the manufacturing of Schottky diodes is not compatible with the CMOS process, requires additional steps that can cause increased costs, and cannot be integrated into mainstream CMOS integrated circuits. As CMOS technology grows, diode-connected transistors are gradually replacing Schottky diodes; however, the *V_TH_* of diode-connected transistors is still higher than that of Schottky diodes. To address these problems, a series of compensation techniques have been proposed. Static *V_TH_* compensation techniques were proposed, to reduce forward losses, and to lower the forward on-resistance to obtain better forward bias; however, a too-low *V_TH_* will cause larger reverse leakage current. Active compensation techniques have been introduced, to reduce *V_TH_* for forward bias, and to increase *V_TH_* for reverse bias, while reducing forward and reverse losses [76,77,78].

As shown in Figure 13, the cross-coupled differential drive rectifier (CCDR) topology is similar to the cross-coupled design of the CP, except that in the rectifier form there is no clock, but instead a differential-mode signal. The CCDR can reduce the ON resistance in forward bias and the leakage current in reverse bias, and has been widely used because of its low-voltage and auto-switching characteristics: however, large leakage currents are still generated, due to the simultaneous conduction of PMOS and NMOS during the transition phase, which can lead to a narrow-peak PCE, even though the cross-coupled configuration has higher PCE, compared to the Dickson structure.

To widen the operating range of RF–DC converters, reducing leakage current is the key. A self-biasing scheme was proposed in [79], to limit reverse leakage at high-input power levels, which improved the input power range by more than 50%, compared to the conventional cross-coupled rectifier: however, the minimum input RF signal still faced the limiting of the *V_TH_*. To compensate the *V_TH_*, Gharehbaghi et al. presented a self-calibration technique in UHF rectifiers, while avoiding an efficiency drop due to the wireless power variation [80]: this technique achieved near-constant PCE values under varying conditions, but the maximum PCE was only 34%.

In addition, Khan et al. have combined two rectifiers, and have proposed a reconfigurable power converter using dual path, as shown in Figure 14, switching the series path at low power and the parallel path at high power, to thus maintain high PCE over a wide input power range [81].

### 5.2. Maximum Power Extraction

Due to the nonlinearity of the rectifier in RF systems, extracting the maximum power from the RF energy can be done using an adaptive impedance matching network method, which can be used to compensate for variations in the input power or load [82], as shown in Figure 15a.

For systems that use the Dickson rectifier, the impedance of the rectifier can be matched to the load by varying the number of the rectifier’s stages—namely, the reconfigurable method [83]. If the RF–DC rectifier is followed by a DC–DC converter, the energy transfer can also be maximized by the MPPT method of the DC–DC converter [84], as shown in Figure 15b.

## 6. Multi-Source Energy Harvesting

To enhance system reliability, energy harvesting systems employing the cooperative collection of multiple energy sources are being increasingly studied.

### 6.1. DC–AC Hybrid Energy Source Acquisition

The block diagram of a conventional multi-energy harvesting system is shown in Figure 16. All the energy enters the combiner after a single conversion. In addition, the role of an optional DC–DC converter after the combiner is to adjust the output voltage needed by the load.

Combiners and DC–DC converters typically achieve conversion efficiencies higher than 80%, whereas AC– DC converters have efficiencies distributed between 20% and 75%, which are strongly influenced by the input power level and load conditions; the system efficiency is greatly impacted by the rectifier [30].

Liu et al. have proposed a new power combining method, where the AC–DC converter is used as a combiner, and the AC energy is converted only once before it reaches the load [30]. The PCE is improved by combining the energy in the rectification stage, as shown in Figure 17. After combining the functions of the AC–DC converter and the combiner, the system efficiency can reach more than 60%.

### 6.2. Architecture of the Combiner

The simplest combined energy structure is shown in Figure 18a, where each energy source is connected to the DC–DC converter through a diode [85,86]: this approach provides a highly modular solution, and allows arbitrary expansion of the input. The drawback is that only the highest energy source can be captured, and the energy from the other ports will be wasted. Another point is that the voltage drop of the diode is also an important factor limiting the efficiency. As in Figure 18b, the use of switches instead of diodes in [35,87] reduces conduction losses, but other issues are not addressed.

The switched-inductor converter structure can be used in combiners, to enable simultaneous multi-source acquisition. A single-inductor multiple-input multiple-output (SIMIMO) converter, with four harvesters, one battery input, and three outputs of 1.8 V, 3.3 V, and one battery, respectively, is proposed in [88] for the MPPT operation, and multiple inputs can be acquired simultaneously. The structure of the inductor-based combiner is shown in Figure 19a.

The capacitor-based structure shown in Figure 19b allows for superior integration performance compared to inductor-based. Estrada-López et al. use capacitor-based structure to achieve energy harvesting from a maximum of two input sources with full integration and maximum power output [31]; in addition, the inputs are sorted according to power level, making it possible to automatically select the two inputs with the highest power to be stored.

## 7. Future Directions and Recommendations

Due to the high energy density and the most mature technology, solar energy is the most widely used. In the design, it is necessary to address the influencing factors that will reduce the system efficiency, such as partial shadows and environmental changes, while finding the balance of system power consumption and MPPT tracking accuracy. Thermal energy harvesting is so low in energy density that the circuit needs to consider self-starting issues and losses in the energy path to be minimized. With respect to DC–DC, the charge pump structure is a promising approach. Based on the charge pump, gate biasing and body biasing focus on lowering the threshold voltage, thus reducing the leakage current, and thereby increasing the efficiency or decreasing the supply voltage, etc. The additional clocking and auxiliary circuitry required to solve these problems increase the overall power consumption of the circuit. The cross-coupled structure can integrate gate bias and body bias techniques to further improve circuit performance, and the advantage of clock boost is the reduction in rise time. Overall, the peak efficiency of the DC–DC converter can reach over 80%. Owing to its high energy density, piezoelectric energy has become the most widely used micro energy, after solar energy, in energy harvesting systems. The efficiency of piezoelectric rectification circuits is extensively measured by the MOPIR; various circuits based on FBR improvements—mainly active rectifier and inductor-based and capacitor-based—have shown in the literature that the MOPIR can reach more than 9. The SSHI is extensively used as a nonlinear method to offset the internal capacitance of the piezoelectric transducer. The advantage of the SSHC is the small area occupied and the possibility of integration. Piezoelectric energy harvesting also requires attention to the effect of different loads on the circuit efficiency. In contrast to other piezoelectric rectification methods, the efficiency of the SECE does not depend on the load. In addition, the design of the piezoelectric energy harvesting circuit also needs to consider the MPPT issue. RF energy input power is weak, and vulnerable to the influence of transmitter distance. The problem of generally narrow peak efficiency also needs to be solved. RF rectification is usually optimized in the CCDR structure, and the peak efficiency can reach more than 75%. Multiple energy co-acquisition is an essential solution for practical application, which needs to be selected with the specific application scenario for suitable micro energy, combined with MPPT technology for energy classification and combination.

Among the MPPT methods applicable to low-power energy harvesting, the P&O method relies on voltage, current sensors, and digital control units. The higher accuracy comes at the cost of higher power consumption. Although FOCV has lower power consumption, it inevitably generates losses during circuit disconnection, and the tracking accuracy is not very high. The recent NN method is only used in high-power systems such as solar energy harvesting, due to slow convergence and high power consumption—however, it has high adaptability to energy sources. To sum up, the more complex but highly accurate MPPT method is suitable for systems with high power levels (mW range), while for systems with low power (µW level), a simple MPPT method with sufficient accuracy may be one of the best implementation options.

WSNs are in sleep mode more than 99% of the time, so their average power consumption is about a few tens of µW, or even less; therefore, when placed in harsh environments, energy management circuits that can efficiently harvest energy from less than 15 µW of ambient energy to power sensors are effective [89]. The miniaturization and off-grid nature of WSN nodes also imposes size requirements on energy management systems. Therefore, the main challenges in the ultra-low-power energy harvesting design scenario include PCE and the circuit size. Reducing the starting voltage is an important means of improving efficiency: the lower the start-up voltage, the more energy can be captured; the higher the efficiency, the more residual energy can be transferred to the load. MPPT is an important way to improve PCE. In the future, it may be possible to allow more complex but effective control methods to be applied to energy harvesting technologies, by optimizing structure and efficiency.

## 8. Conclusions

This review paper presents a number of schemes for low-power, low-voltage circuits for micro energy harvesting, which indicates that the goal of harvesting ambient energy to supply IoT nodes has been fully developed: however, there are still many challenges to be solved in practical applications. This paper describes the characteristics of the four types of energy, as well as the circuit components, in detail; combined energy, as a method to ensure system reliability, is also presented. Efficiency and circuit size constrain the development of micro energy harvesting: improved efficiency can be achieved by more accurate control circuits and smaller losses; to make the circuit size smaller and easier to integrate on-chip, capacitor-based solutions are more promising than inductors.

## Figures and Tables

**Figure 1 micromachines-13-02222-f001:**
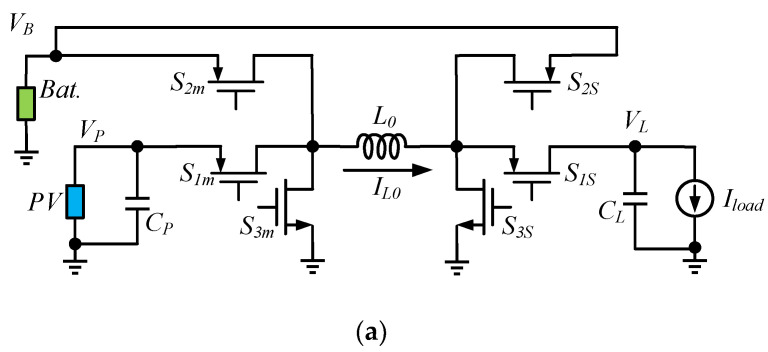
(**a**) 2P6S converter, (**b**) 2P3S converter, (**c**) 1P3S converter.

**Figure 2 micromachines-13-02222-f002:**
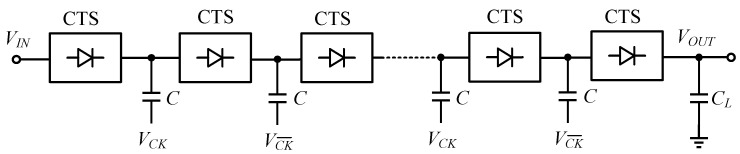
Simplified scheme of a linear Dickson charge pump.

**Figure 3 micromachines-13-02222-f003:**
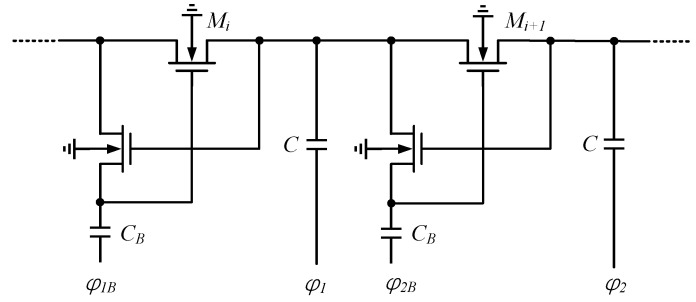
Gate biasing CP.

**Figure 4 micromachines-13-02222-f004:**
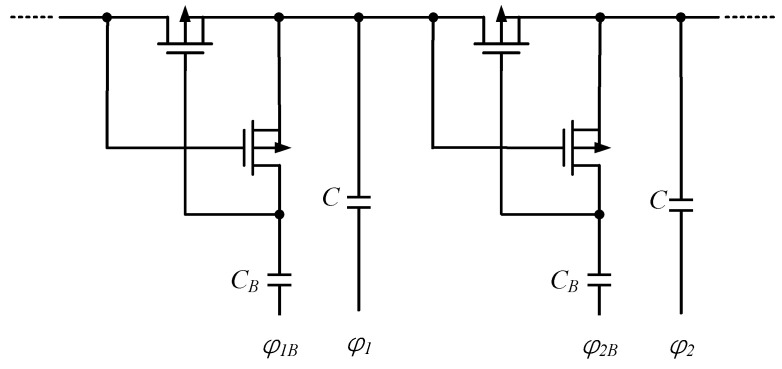
Body biasing CP.

**Figure 5 micromachines-13-02222-f005:**
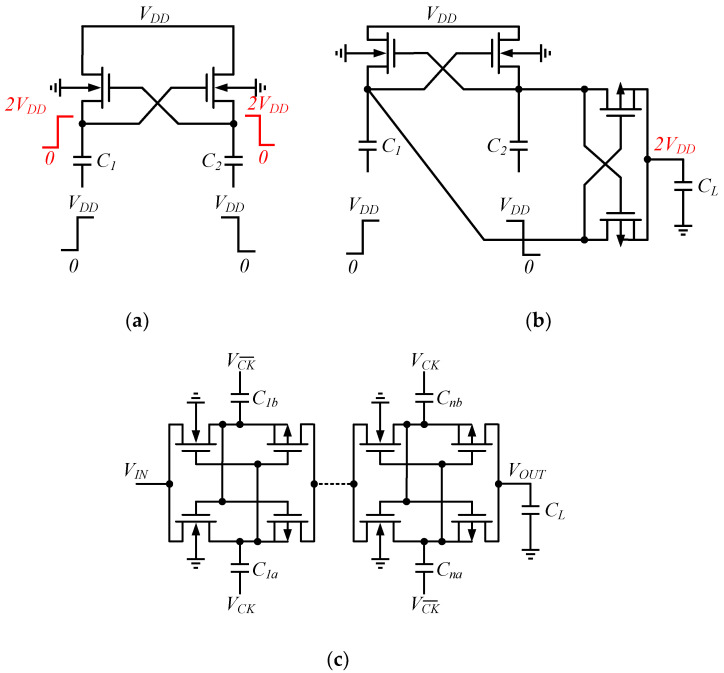
Nakagome’s cell scheme: (**a**) without PMOS switches; (**b**) with PMOS switches; (**c**) a cross-coupled charge pump.

**Figure 6 micromachines-13-02222-f006:**
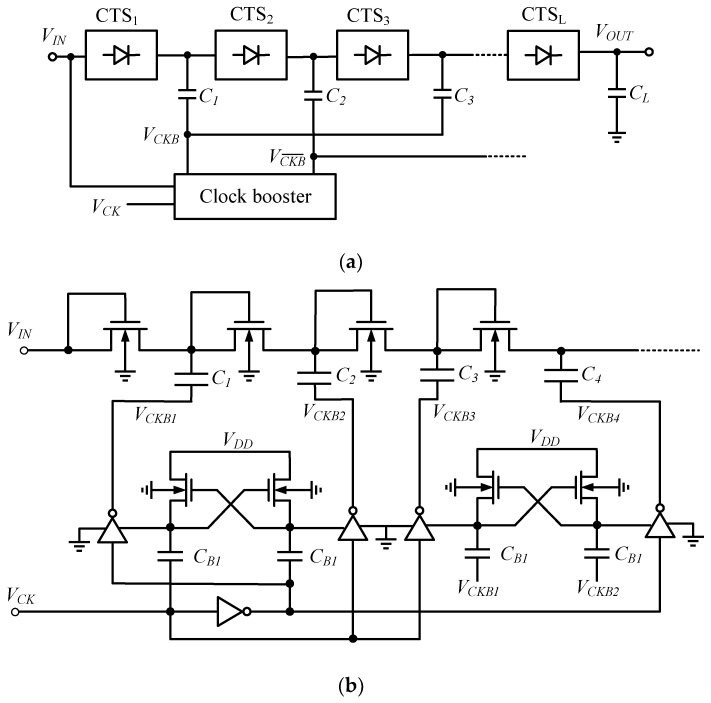
Clock-boosted CP: (**a**) simplified scheme; (**b**) multi-stage scheme.

**Figure 7 micromachines-13-02222-f007:**
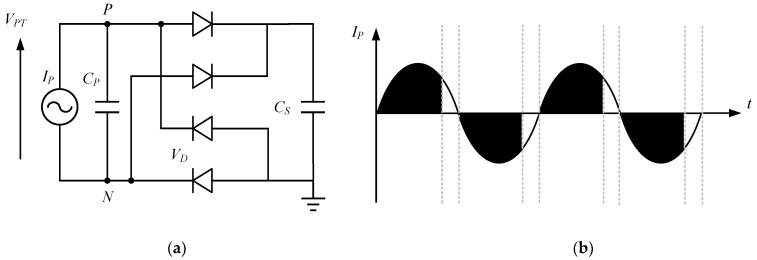
FBR (**a**) structure, (**b**) waveforms.

**Figure 8 micromachines-13-02222-f008:**
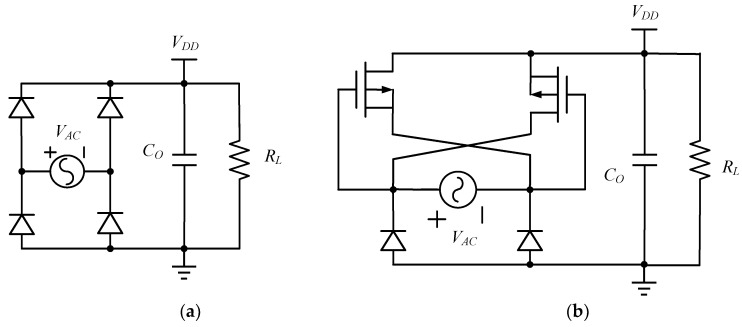
(**a**) FBR, (**b**) rectifier with cross-coupled PMOS.

**Figure 9 micromachines-13-02222-f009:**
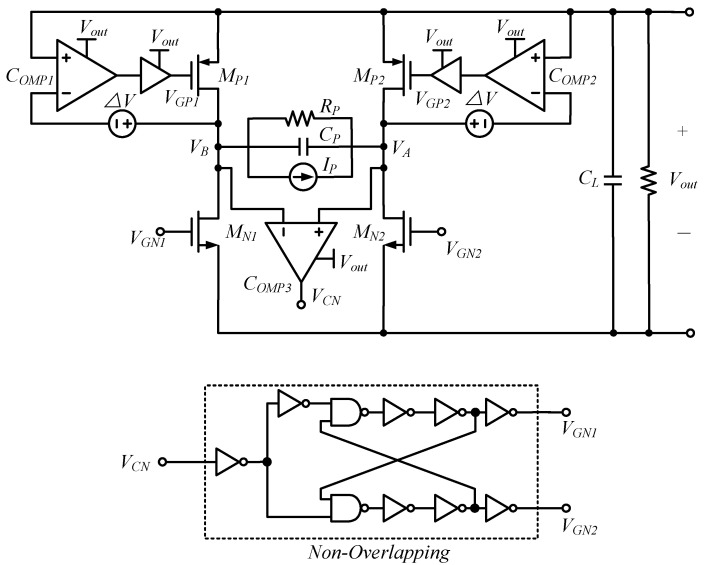
Active rectifier with non-overlapping control.

**Figure 10 micromachines-13-02222-f010:**
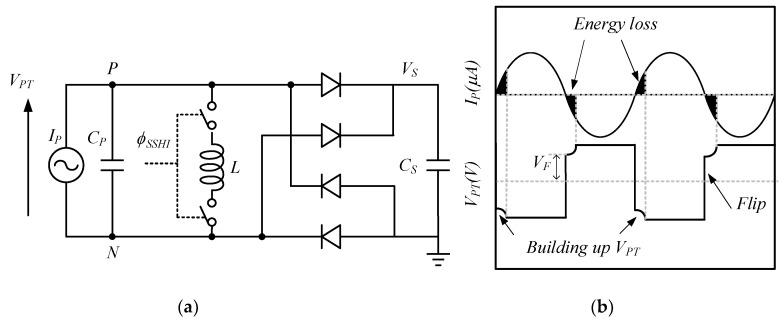
SSHI (**a**) structure, (**b**) waveforms [55].

**Figure 11 micromachines-13-02222-f011:**
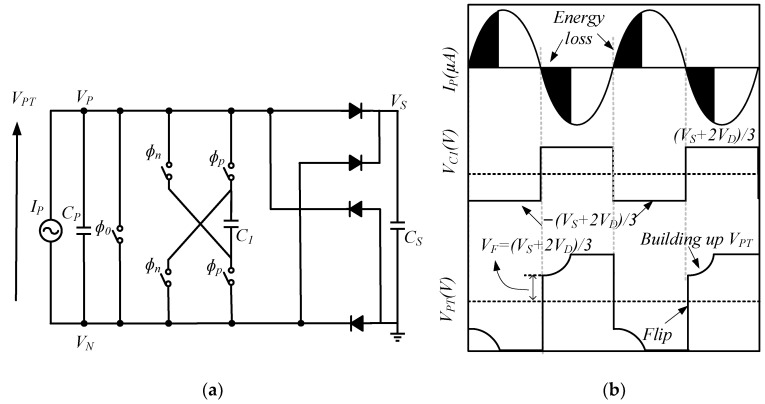
SSHC (**a**) with one charge-swap capacitor, (**b**) waveforms [55].

**Figure 12 micromachines-13-02222-f012:**
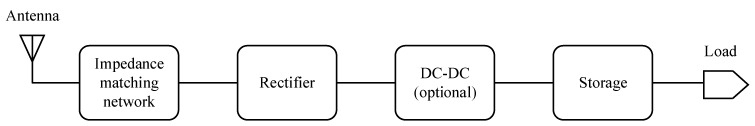
The architecture of the RFEH system.

**Figure 13 micromachines-13-02222-f013:**
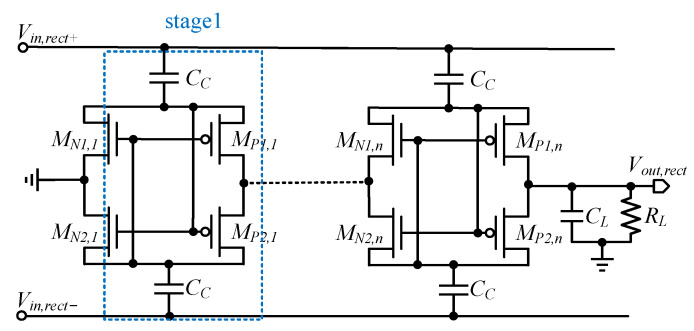
Cross-coupled differential drive rectifier (CCDR) [74].

**Figure 14 micromachines-13-02222-f014:**
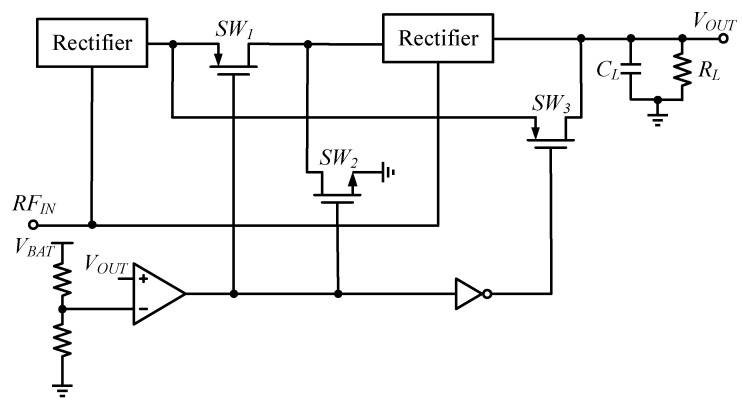
The reconfigurable RF–DC power converter.

**Figure 15 micromachines-13-02222-f015:**
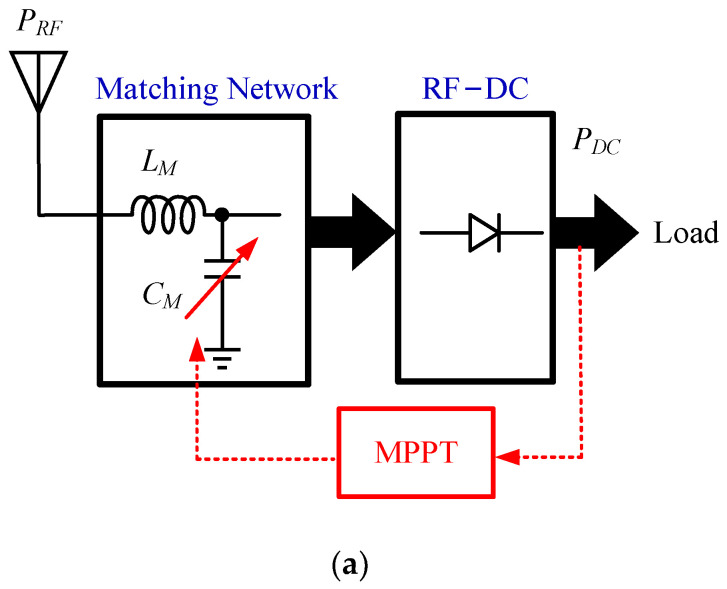
MPPT for RF–DC: (**a**) with adaptive matching network; (**b**) with cascaded boost converter for adaptive load control [30].

**Figure 16 micromachines-13-02222-f016:**
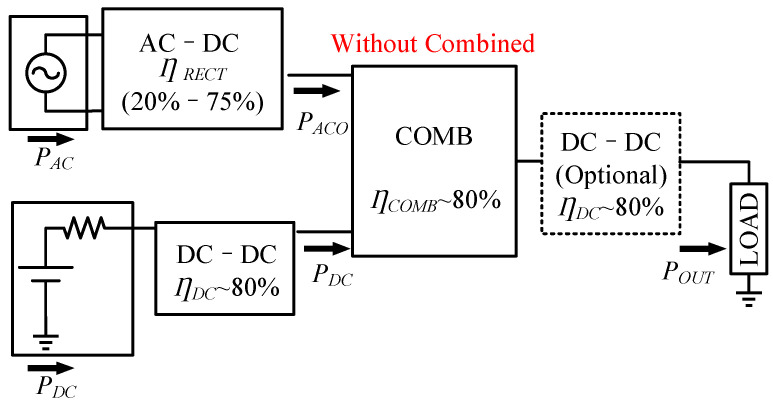
Conventional combiner system structure with mixed AC and DC harvesting.

**Figure 17 micromachines-13-02222-f017:**
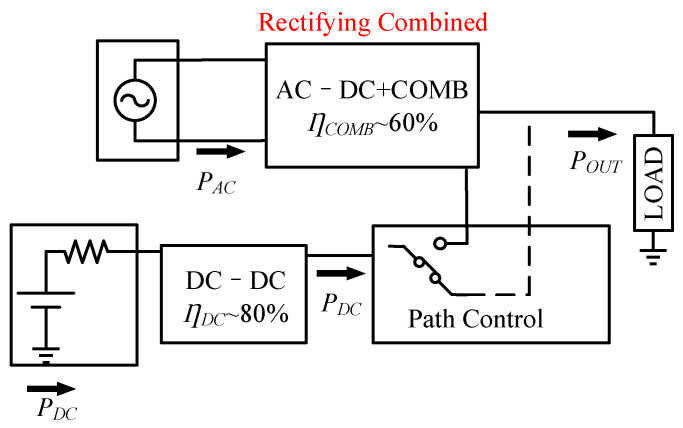
Improved mixed AC and DC harvesting system.

**Figure 18 micromachines-13-02222-f018:**
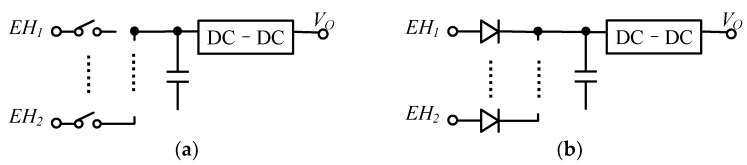
Architecture of energy combining (**a**) diode-connected, (**b**) switch-connected.

**Figure 19 micromachines-13-02222-f019:**
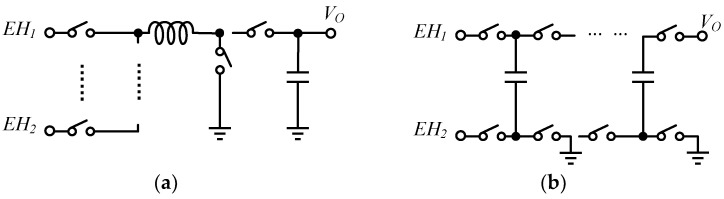
Architecture of energy combining (**a**) inductor-based, (**b**) capacitor-based.

**Table 1 micromachines-13-02222-t001:** Classifications of Energy Sources.

Type	PVC	TEG	PEH	WEH
Voltage	High	Low	High	High
Current	Low	High	Low	Low
Type	DC	DC	AC	AC
Drive Capability	General	Strong	General	Weak

DC, direct current; AC, alternating current.

## Data Availability

Not applicable.

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
