# Peer review of "A Review of Converter Circuits for Ambient Micro Energy Harvesting"

_micromachines, 2022, doi:10.3390/mi13122222_

Round 1
Reviewer 1 Report
Authors have presented a review of converter circuits for ambient micro-energy harvesting. The work is useful. Following comments will be helpful to further improve the manuscript.
Figures are smaller in size. The size should be increased for better visibility to an adequate size.
It would be great to add more details/comments on potential applications to section 7 “Future Directions and Recommendations”
Check the manuscript carefully for grammatical errors and/or typos.
Overall, the work is useful and is well presented.
Reviewer 2 Report
The review provided in this paper is interesting and actual. However, please notice comments given below in order to upgrade your paper.
· The converter losses mentioned in line 107 are semiconductor losses because the converter losses also comprise the losses of the passive components such as the inductor losses.
· The sentence in line 116 is not clear, it shouldn’t start with [8]. Please correct and check entire text.
· Symbols should be written in italic in Figs and text. For example, VTH in line 180. Please correct entire text.
· P&O should be described in the first occurrence in the text.
· Some sort of comparison of the considered methods based on simulations (e.g. Matlab Simulink) or in better case experiments should be shown.
Reviewer 3 Report
See attached.

Round 2
Reviewer 2 Report
I think the paper is now ready for the publication.
Reviewer 3 Report
accept